# Thermal annihilation of photo-induced radicals following dynamic nuclear polarization to produce transportable frozen hyperpolarized $^{13}$C-substrates

Andrea Capozzi[1], Tian Cheng[1], Giovanni Boero[2], Christophe Roussel[3] & Arnaud Comment[1,4]

Hyperpolarization via dynamic nuclear polarization (DNP) is pivotal for boosting magnetic resonance imaging (MRI) sensitivity and dissolution DNP can be used to perform *in vivo* real-time $^{13}$C MRI. The type of applications is however limited by the relatively fast decay time of the hyperpolarized spin state together with the constraint of having to polarize the $^{13}$C spins in a dedicated apparatus nearby but separated from the MRI magnet. We herein demonstrate that by polarizing $^{13}$C with photo-induced radicals, which can be subsequently annihilated using a thermalization process that maintains the sample temperature below its melting point, hyperpolarized $^{13}$C-substrates can be extracted from the DNP apparatus in the solid form, while maintaining the enhanced $^{13}$C polarization. The melting procedure necessary to transform the frozen solid into an injectable solution containing the hyperpolarized $^{13}$C-substrates can therefore be performed *ex situ*, up to several hours after extraction and storage of the polarized solid.

[1] Institute of Physics of Biological Systems, École Polytechnique Fédérale de Lausanne, CH-1015 Lausanne, Switzerland. [2] Institute of Microengineering, École Polytechnique Fédérale de Lausanne, CH-1015 Lausanne, Switzerland. [3] Section of Chemistry and Chemical Engineering, Institute of Chemical Sciences and Engineering, École Polytechnique Fédérale de Lausanne, CH-1015 Lausanne, Switzerland. [4] General Electric Healthcare, Pollards Wood, Nightingales Lane, Chalfont St Giles, Buckinghamshire HP8 4SP, UK. Correspondence and requests for materials should be addressed to A.C. (email: arnaud.comment@ge.com).

Hyperpolarized $^{13}$C magnetic resonance imaging (MRI) is one among several molecular imaging techniques proposed in the recent years to detect biochemical changes *in vivo*[1,2]. Most imaging modalities, including MRI, computed tomography, positron emission tomography, single photon emission computed tomography, ultrasound and a variety of optical imaging methods can be adapted to reveal insights into cellular function[3]. MRI can provide direct biochemical information via the spectroscopic dimension of nuclear magnetic resonance (NMR), allowing simultaneous acquisition of signals from a substrate and its metabolic products, hence yielding true metabolic imaging. The relatively low sensitivity of NMR can be circumvented using hyperpolarization techniques such as spin-exchange optical pumping for gases, and dissolution dynamic nuclear polarization (DNP), parahydrogen-induced polarization, as well as the so-called 'brute force' method for liquids[4–6]. In the context of metabolic imaging, $^{13}$C is the most adapted nucleus because of its ubiquitous presence in the vast majority of biomolecules, its large chemical shift dispersion allowing to easily differentiate the various species, its low natural abundance and its relatively long longitudinal relaxation time at specific molecular positions such as in a carboxyl group[1,7–11]. Parahydrogen-induced polarization was the first hyperpolarization technique proposed for *in vivo* $^{13}$C MRI[12], but dissolution DNP became more popular because of its versatility for applications in biomedical imaging[13,14].

The origin of the NMR signal being the electromotive force generated by the magnetic moment of the nuclear spins precessing inside a radiofrequency (rf) coil, the main physical reason behind the low sensitivity of NMR is the small magnitude of the nuclear spin magnetic moments combined with their low polarization[15]. The strength of the electromotive force can be enhanced by increasing the amplitude of the nuclear spin polarization, which is defined, for a spin-½ nucleus such as $^{13}$C, by $P_n = \tan h(h\nu/2k_BT)$, where $h$ is the Planck constant, $\nu$ the resonance frequency of the nuclear spins of interest, $k_B$ the Boltzmann constant and $T$ the temperature of the environment surrounding the nuclear spins. $P_n$ will be greater at larger magnetic field, but it can additionally be increased by decreasing the temperature. For medical applications, $T$ is fixed by the organism's temperature and can therefore not be adjusted. It is however possible to reduce the nuclear spin temperature within warm molecules. Indeed, some nuclear spins are only weakly coupled to their environment and a highly out-of-thermal-equilibrium spin state can therefore be maintained for a substantial amount of time after having precooled the molecules. This had already been demonstrated in 1958 by Abragam and Proctor[16]. Low $^{13}$C spin temperatures can be achieved using the brute force approach as was recently proposed by Hirsch *et al.*[17], but the polarization times are prohibitively long and the polarization levels are limited.

For polarizing nuclear spins in the solid state, the DNP method originally introduced by Overhauser has been shown to be the most effective method to reach near-unity polarization at low temperature[18,19]. The idea is to take advantage of the large magnetic moment of the electron spins, about 2,400 times larger than the $^{13}$C magnetic moment. However, because of the high NMR relaxivity of unpaired electron spins at temperatures above liquid helium and in small magnetic field[17,18], it is not possible to extract the frozen solid from the high-field and low-temperature environment required for solid-state DNP without losing most of the nuclear spin polarization. For this reason, an *in situ* dissolution step was designed by Ardenkjaer-Larsen *et al.*[14] and it was demonstrated that liquid-state $^{13}$C polarization enhancement larger than $10^4$ could be achieved. The fact that the $^{13}$C spins are extracted in a liquid form has however some

disadvantages: it cannot be transported very far from its production site because of the relatively short relaxation time, typically shorter than a minute (this time can be extended in some specific molecules through the formation of a so-called long-lived state[20,21], but no decisive application for *in vivo* metabolic imaging has been proposed to date[22]); moreover, the $^{13}$C-subtrate concentration in the produced solution is limited because of the dilution factor determined by the minimum amount of solvent required for a complete dissolution of the frozen solid. It is noteworthy that a dissolution-DNP method allowing the preparation of transportable hyperpolarized metabolites was recently proposed but it requires the use of non-biocompatible solvents and is therefore not suitable for *in vivo* applications[23].

In this study, we take advantage of the non-persistent nature of specific photo-induced radicals to produce hyperpolarized $^{13}$C-substrates that can be extracted from the DNP apparatus without the need for a dissolution process. The frozen solid containing hyperpolarized $^{13}$C-substrates can consequently be melted at a later time, in a remote location, and this method also extends the potential applications of hyperpolarized $^{13}$C MRI to biomolecules with low solubility.

## Results

**Thermalization method to annihilate photo-induced radicals.** It was recently shown that photo-induced radicals can be used to polarize pyruvic acid (PA) and other molecules[24,25]. These radicals, formed at 77 K by ultraviolet-light irradiation of frozen aliquots containing PA, do not persist when brought to room temperature. Using electron spin resonance (ESR), we determined that the radicals are annihilated if the sample temperature increases above $190 \pm 2$ K (Fig. 1). This observation is in agreement with previously reported measurements[26]. As the fusion temperature of frozen PA is around 285 K, there is a 95 K temperature window within which the radicals do not persist while the sample is still in its solid state. Based on this intrinsic property, we designed a rapid thermalization method for photo-irradiated frozen samples containing PA. It allows annihilating the radicals while maintaining the $^{13}$C polarization enhanced via low-temperature DNP without melting the frozen samples. To raise the temperature of a 2 mm diameter sphere

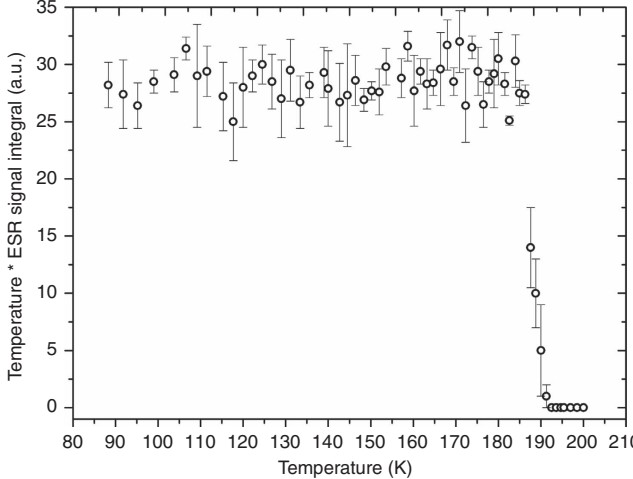

**Figure 1 | Annihilation temperature of photo-induced radicals.** ESR signal integral multiplied by temperature as a function of temperature. Each ESR measurement is the average of three single scans (the error bar represents the s.d.).

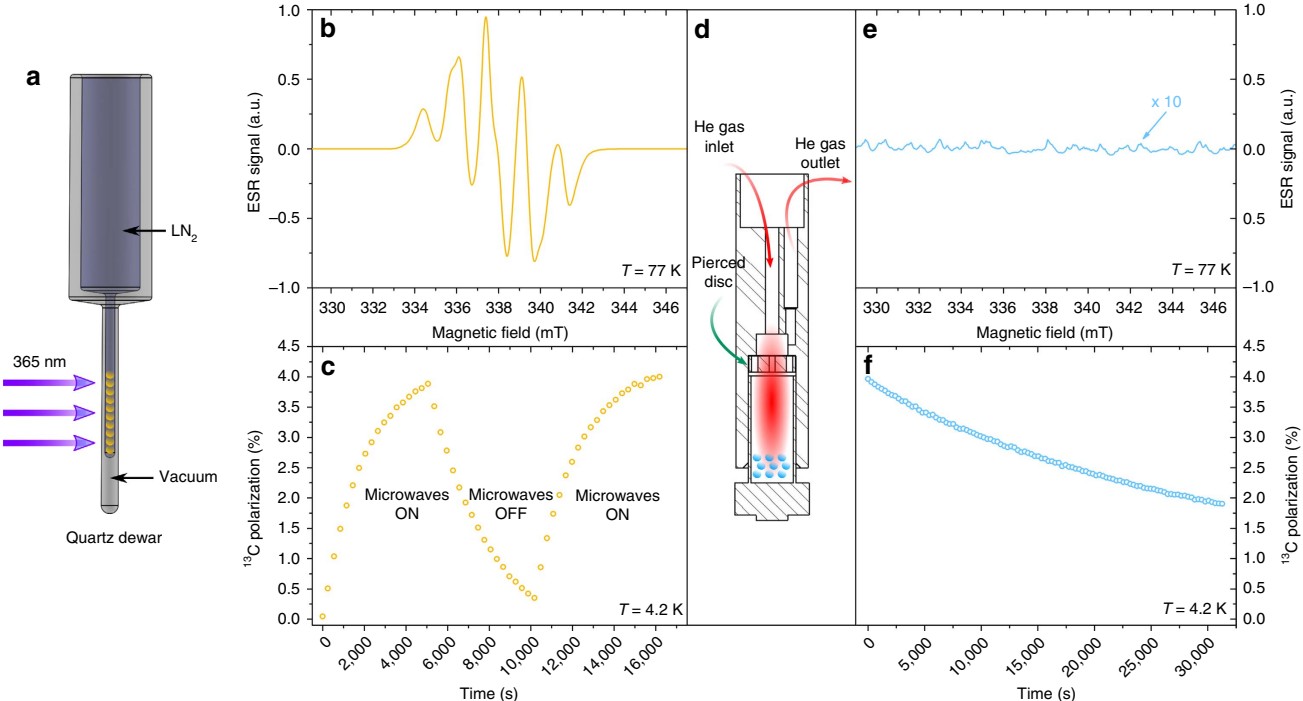

**Figure 2 | Rapid thermalization method.** (**a**) Sample preparation: frozen beads of [1-$^{13}$C]PA:H$_2$O 1:1 (v/v) placed in a quartz dewar filled with liquid nitrogen are irradiated with ultraviolet light (365 nm) for 1h. The orange colour indicates the presence of radicals. (**b**) X-band ESR spectrum collected at 77 K in a single ultraviolet-irradiated frozen bead. (**c**) $^{13}$C polarization time evolution measured by $^{13}$C NMR at 7 T and 4.2 K in ten ultraviolet-irradiated frozen beads during an ON–OFF–ON microwave cycle. (**d**) Sketch of the bottom part of the thermalization insert sealed onto the sample cup. The compressed helium gas inlet (3 mm diameter) and outlet (2 mm diameter) are sketched along with the warm helium gas profile (red cloud) inside the sample cup together with the frozen beads (the cyan colour indicates that the ultraviolet-induced radicals have been annihilated). (**e**) X-band ESR spectrum measured at 77 K in all ten ultraviolet-irradiated beads recovered from the sample cup after thermalization and $^{13}$C NMR longitudinal relaxation measurement. (**f**) $^{13}$C NMR longitudinal relaxation measurement performed at 4.2 K and 7 T following thermalization.

composed of a frozen PA and water mixture (4 μl of 1:1 (v/v) [1-$^{13}$C]PA:H$_2$O) from 1 to 190 K, a minimum energy of 1 J is necessary and it would melt if > 2.5 J was delivered to the sphere (see Supplementary Note 1). Knowing that the thermal capacity of helium at room temperature is about 0.85 J l$^{-1}$ K$^{-1}$, we anticipated that a couple liters of helium gas should be sufficient to bring the temperature of a sample composed of 4 μl spherical beads to within the above-defined 95 K window. We empirically determined that a volume of 1.8 l dispensed during 3 s at 2.5 bar maintains the solid beads intact and is sufficient to thermalize them above the annihilation temperature to eliminate all ultraviolet-induced radicals. This was confirmed by measuring and comparing the X-band ESR signal from ultraviolet-irradiated beads (Fig. 2a) before and after thermalization (Fig. 2b,e). The prominent advantage of using helium gas as thermalization fluid is that no impurities are introduced into the cryogenic environment, even in case of a mechanical failure leading to a leakage of the thermalization fluid.

**DNP and solid-state $^{13}$C NMR at 4.2 K.** A specific experiment was designed to demonstrate that the $^{13}$C polarization is conserved through the proposed rapid thermalization method (Fig. 2c,d,f). A $^{13}$C NMR probe placed inside the polarizer cryostat was used for this purpose. All measurements were performed at 4.2 K to ensure that the helium bath, as well as the NMR probe, would stay at constant temperature during the whole procedure, including during and after the pressurization period required to maintain the cryostat at atmospheric pressure while introducing the insert designed for sample thermalization (Fig. 2d). A first set of $^{13}$C DNP build-up and longitudinal

relaxation measurements were performed to determine the dynamic parameters of the sample (Fig. 2c). The maximum $^{13}$C polarization was found to be 4 ± 0.2%. A second build-up measurement was performed at the end of the relaxation measurement and once the $^{13}$C polarization had reached its maximum value, the sample was raised above the liquid helium bath prior to initiate the thermalization process. Following thermalization, the sample was moved back to its original position inside the coil immersed into liquid helium and another relaxation measurement was recorded (Fig. 2f).

The results demonstrated that the thermalization process does not significantly affect the $^{13}$C polarization level achieved following microwave irradiation since the NMR signal intensity measured just after thermalization was around 95% of the value recorded before thermalization. It must however be borne in mind that before radical annihilation, paramagnetic relaxation will deteriorate the $^{13}$C polarization if the magnetic field is too low or the sample temperature stays significantly higher than 4.2 K for an extended period of time. It is therefore necessary to avoid moving the sample too far above the isocentre of the superconducting magnet and to rapidly raise the temperature above the annihilation temperature. These requirements are identical to those pertaining to dissolution DNP[14]. Following thermalization, the $^{13}$C $T_1$ increased by a factor larger than 18, from 3,200 ± 200 s to 58,000 ± 1,000 s (Fig. 2c,f). Using cold pressurized helium gas, the frozen beads were then rapidly (in about 1 s) extracted and transferred into a quartz dewar filled with liquid nitrogen for subsequent ESR measurements, to confirm the absence of radicals (Fig. 3). Control experiments using the same extraction procedure were performed on beads that did not go through the thermalization process to verify

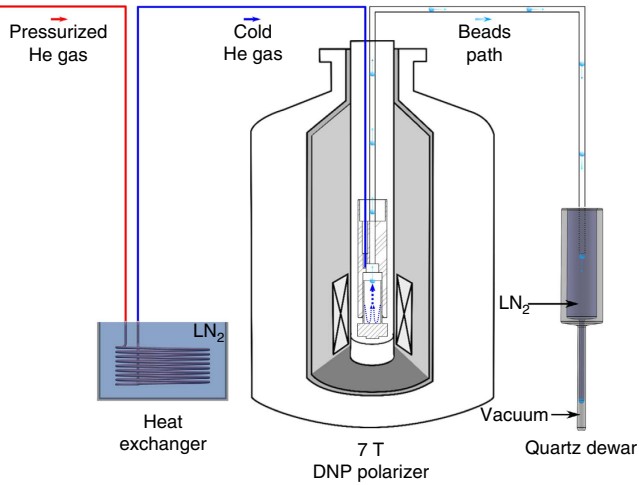

**Figure 3 | Solid sample extraction setup.** Setup used to extract the frozen beads from the DNP polarizer. The pressurized helium gas path is sketched from left to right: room-temperature helium gas (5 bar) is cooled through a heat exchanger (copper solenoid plunged in liquid nitrogen); the cold gas enters the extraction insert (only the bottom part is sketched inside the DNP polarizer) through the 2 mm diameter inlet and the frozen beads are pushed out of the sample cup through the 3 mm outlet; the extracted beads are recovered inside a quartz dewar filled with liquid nitrogen located next to the DNP polarizer.

that the radicals were not fortuitously scavenged along the extraction path.

**Extraction and *ex situ* melting of polarized $^{13}C$ subtrates.** The elimination of the radicals in the solid state alleviates the crucial requirement for dissolution to take place inside the high-field and low-temperature environment of the polarizer. To demonstrate that it is indeed possible to perform the solid-to-liquid transformation outside of the polarizer after extraction of the frozen beads without considerable reduction of the $^{13}C$ polarization, an additional set of experiments was performed. Following DNP, thermalization and extraction of the polarized solid sample, the beads were directly melted inside a glass container filled with warm $D_2O$ and placed in the centre of a 100 mT permanent magnet adjacent to the polarizer (Fig. 4). The resulting solution was then transported into the fringe field (30 G) of a 9.4 T 31 cm horizontal bore MRI scanner, located about 10 m away from the polarizer and rapidly introduced into a custom-designed injection pump located inside the 9.4 T magnet bore. This injection pump is designed for rodent experiments and equipped with a $^{13}C$ NMR probe (see ref. 27 for details). The hyperpolarized $^{13}C$ signal was measured 15 s after sample melting and the enhancement determined by comparison with the thermally polarized $^{13}C$ signal recorded on the same solution after complete relaxation of the $^{13}C$ spins (see Methods). The liquid-state $^{13}C$ polarization was found to be $3 \pm 0.3\%$. A series of control experiments confirmed that when omitting the thermalization process, the exact same protocol leads to no detectable hyperpolarized $^{13}C$ signal.

The final set of experiments consisted in repeating the same protocol following DNP at 1 K instead of 4.2 K, to evaluate the maximum achievable $^{13}C$ polarization (Fig. 4b,d,f). After 3 h of microwave irradiation, the solid-state $^{13}C$ polarization reached $12.0 \pm 0.5\%$. The enhancement measured at 9.4 T after thermalization, extraction, *ex situ* melting and transfer into the injection pump was found to be $10,400 \pm 900$, corresponding to a

$^{13}C$ liquid-state polarization of $8.3 \pm 0.7\%$. With an estimated low-field liquid-state $^{13}C$ $T_1$ of 40 s, the extrapolated $^{13}C$ polarization just after melting was essentially identical to the measured solid-state polarization. In other words, the extraction and *ex situ* melting processes are nearly lossless. A mobile sample holder such as the one developed by Hirsch *et al.*[17] would allow transporting the solid sample to a remote location prior to melting. It is noteworthy that it has also been demonstrated that hyperpolarized $^{129}Xe$ can be stored for an extended period of time and transported in the solid form at cryogenic temperature[28,29].

## Discussion

The liquid-state $^{13}C$ polarization obtained in the present work is severalfold lower than the maximum values obtained by dissolution DNP using the same polarizer and trityl radicals[30]. The narrower ESR line width of trityl radicals as compared to photo-induced radicals is expected to lead to higher $^{13}C$ polarization[25], but the main reason for the lower polarization is the sub-optimal radical concentration, which also explains that the build-up time constant is larger than in an optimized PA sample prepared with trityl radical. Preliminary results show that this concentration could be improved by using a broadband ultraviolet light source, but a complete optical study is beyond the scope of this manuscript. Modulating the microwave frequency or performing $^1H$-$^{13}C$ cross-polarization could also improve the polarization level and reduce the build-up time[31,32], but these methods require additional hardware. The herein presented technique can be used to polarize other $^{13}C$-compounds as well as other nuclei following the method proposed by Capozzi *et al.*[24]. As other alpha-keto acids exhibit similar photochemical behaviour as PA[33], we expect to be able to extend this technique to other biomolecules in the near future.

In conclusion, by coupling the developed rapid thermalization method and low-temperature DNP with photo-induced radicals, it is possible to produce highly polarized solid consisting of exclusively endogenous substrates for metabolic imaging. This opens the way to the development of methods to prepare hyperpolarized substrates for biomedical applications without using exogenous substances[34].

## Methods

**Sample preparation.** Ten $4.0 \pm 0.5\,\mu l$ droplets of a solution composed of [1-$^{13}C$]PA:$H_2O$ 1:1 (v/v) were pipetted, one by one, inside a synthetic quartz dewar (Wilmad 150 ml Suprasil Dewar Flask type WG-850-B-Q) filled with liquid nitrogen to form ~ 2 mm diameter spherical frozen beads. The beads were ultraviolet-irradiated for 60 min using a 365 nm LED array (LC-L5, Hamamatsu Photonics K.K., Hamamatsu, Japan) as described in former publications[24,25].

**X-band ESR measurements at 77 K.** ESR spectra were recorded at 77 K in frozen beads placed within the tail of the quartz dewar inserted inside the resonant cavity of a commercial bench-top X-band spectrometer (Miniscope 400, Magnettech, Berlin, Germany). The radical concentration was measured following previously described methods[24]. In agreement with previously published results[24], it was determined to be $27 \pm 1$ mM after ultraviolet irradiation.

**Annihilation temperature measurements.** To determine the temperature at which the ultraviolet-induced radicals recombine in frozen solutions containing PA, a homemade X-band ESR spectrometer equipped with a half-wavelength coplanar waveguide (CPW) resonator was used (Supplementary Fig. 1). The main apparatus has been previously described[35], but the ESR probe was modified to accommodate for a thermometer (resistance type PT1000) and clamp two frozen beads within the sensitive area of the CPW resonator. The beads were placed on the epoxy-coated rear side of the probe, inside two 2.5 mm diameter hemispherical cavities milled over the centre of the CPW resonator. The beads were clamped using a bridge and two screws made of polyoxymethylene. The PT1000 sensor was glued about 1 mm above the upper cavity and driven by a constant current of 1 mA. The ESR probe was precooled in liquid nitrogen prior to placing the beads inside the hemispherical cavities. A liquid nitrogen cryostat was also precooled and emptied before being placed in the isocentre of the ESR electromagnet.

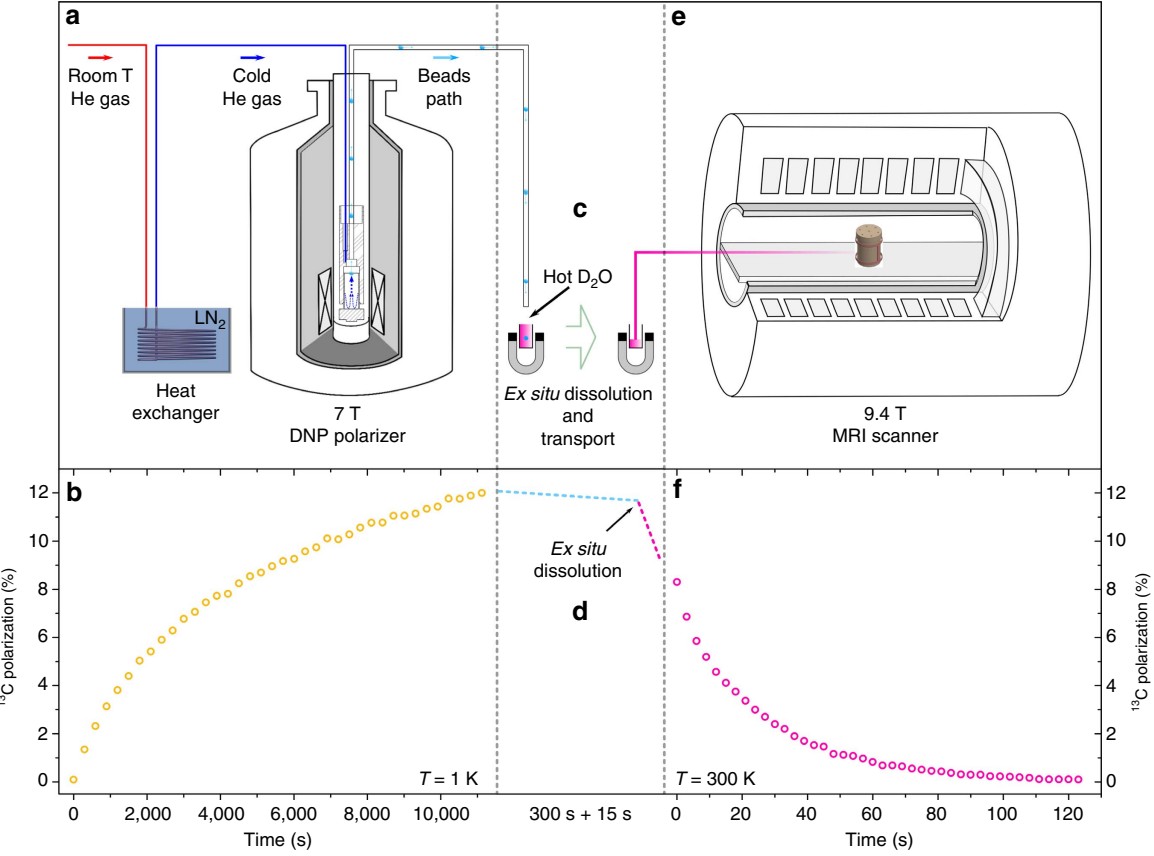

**Figure 4 | *Ex situ* dissolution. (a)** Solid sample extraction procedure as illustrated in Fig. 3. **(b)** [13]C polarization time evolution measured by [13]C NMR at 7 T and 1 K in 10 ultraviolet-irradiated frozen beads; once the [13]C polarization had reached 12 ± 0.5%, the ultraviolet-induced radicals were annihilated using the thermalization process described in the text **(c)**. Schematic representation of the *ex situ* dissolution and transfer of the hyperpolarized [1-[13]C]PA aqueous solution from the side of the polarizer to the custom-designed injection pump placed inside the MRI scanner. **(d)** Estimated [13]C polarization behavior during all intermediate operations: the cyan dotted line represents the decay of the [13]C polarization at 7 T and 4.2 K during the 300 s required to cool down the extraction line; the fuchsia dotted line sketches the liquid-state [13]C longitudinal relaxation after *ex situ* dissolution at 100 mT in hot $D_2O$ during the 15 s needed to transport the hyperpolarized solution in proximity of the 9.4 T MRI scanner. **(e)** Sketch of the injection pump equipped with a [13]C Alderman–Grant NMR coil and placed at the isocentre of the 9.4 T MRI scanner. **(f)** Liquid-state [13]C NMR longitudinal relaxation measurement performed at 9.4 T and room temperature inside the injection pump.

The probe was immediately inserted inside the empty cold cryostat and the ESR measurements were initiated. Natural convection slowly warmed up the probe and the evolution of both the temperature as well as the ESR signal was recorded every 30 s. The measurement was completed in 30 min (Supplementary Fig. 2). The ESR signal integral was then multiplied by the temperature to account for the change in electron spin polarization and reported as a function of temperature (Fig. 1). No ESR signal could be detected above 190 ± 2 K.

**Solid-state DNP measurements at 7 T and 4.2 K.** The ultraviolet-irradiated beads were placed inside a 0.4 ml polytetrafluoroethylene (PTFE) sample cup attached to the end of a dedicated sample holder insert made of fiberglass and then loaded inside a 7 T custom-designed polarizer prefilled with liquid helium (a full description of the DNP polarizer can be found in the original publications)[36,37]. DNP was performed according to previously described methods[38]. The output power of the microwave source was set to 55 mW and the frequency to 196.633 GHz, parameters providing the maximum [13]C DNP enhancement for the ultraviolet-irradiated beads (Supplementary Fig. 3). The [13]C polarization time evolution was monitored using a home-made NMR setup[38], applying a 6° rf pulse at 75.100 MHz every 5 min. Once the maximum polarization was reached, a 30° rf pulse was applied and, after switching off the microwaves and waiting for a sufficiently long time for complete relaxation of the [13]C nuclear spins, an identical 30° rf pulse was applied to measure the reference signal corresponding to the thermal equilibrium [13]C polarization. The DNP enhancement was obtained by computing the ratio between the two NMR signal integrals. A larger flip angle (30°) was necessary to increase the signal-to-noise ratio of the thermal equilibrium signal, to accurately determine the solid-state polarization. It is noteworthy that the observed enhancement and time evolution were similar for samples composed of [1-[13]C]PA:$D_2O$ 1:1 (v/v) and we therefore only focused on samples prepared with $H_2O$. Details about the NMR coil calibration and the rf correction applied to

the time constants can be found in the Supplementary Methods (Supplementary Fig. 4).

**Thermalization process.** To annihilate the ultraviolet-induced radicals without losing the DNP-enhanced [13]C polarization, it is necessary to rapidly increase the temperature of the beads above the annihilation temperature (190 ± 2 K) without melting the beads and while keeping the beads within a large magnetic field at all time. This can be done by using a custom-designed thermalization insert that is compatible with the main insert of a DNP polarizer (see ref. 37). The thermalization insert is introduced inside the polarizer at the end to the DNP process, once the cryostat has been pressurized to atmospheric pressure and the microwave insert has been removed. Before introducing the thermalization insert, the dedicated sample holder is lifted above the liquid helium bath, about 10 cm away from the magnet isocentre. Once the insert is coupled to the sample cup in a leak-tight manner (see Fig. 2d), pressurized room-temperature helium gas is blown onto the beads. At the end of the procedure, the thermalization insert is pulled out of the polarizer and the sample holder is moved back to its previous position, that is, at the isocentre of the polarizer magnet, inside the helium bath. Finally, the microwave insert is placed back into the polarizer to recover the same conditions as during the DNP process, to compare the solid-state [13]C NMR signal before and after the thermalization process. The absence of radicals following thermalization was verified a posteriori by ESR after having rapidly extracted the beads from the polarizer and place them in liquid nitrogen.

The bottom of the thermalization insert was identical to the bottom of the original dissolution insert described in ref. 37, except for the addition of a pierced PTFE disc (thickness: 2 mm; holes diameter: 0.5 mm) designed to maintain the beads inside the 400 mm³ volume of the sample cup and prevent them from occluding the 2 mm diameter gas outlet. The optimal helium gas pressure $P_T$ and thermalization time $\tau_T$ were determined in a separate series of experiments

performed on unlabeled PA samples (ultraviolet-irradiated beads of PA:$H_2O$ 1:1 (v/v)). The optimal thermalization parameters were found to be $P_T = 2.5$ bar and $\tau_T = 3$ s, corresponding to a gas flow of $0.61 s^{-1}$.

**Solid sample extraction procedure.** Dedicated hardware was built to extract the frozen beads from the DNP polarizer (see Fig. 3). The extraction insert was similar to the thermalization insert except for two elements: the pierced disc was removed and the inlet and outlet tubes were swapped. To prevent the beads from melting during the extraction procedure, the outlet PTFE tube was precooled by flushing cold ($\sim 100$ K) pressurized helium gas for 5 min. Helium gas was cooled using a $30$ cm $\times 30$ cm copper spiral heat exchanger plunged in liquid nitrogen (see Fig. 3). The end of the outlet tube was placed inside a synthetic quartz dewar (Wilmad 150 ml Suprasil Dewar Flask type WG-850-B-Q) filled with liquid nitrogen to collect the frozen beads. The coupling of the extraction insert with the sample cup was performed using a procedure identical to the insertion of the thermalization insert. A burst of pressurized helium gas (5 bar) was applied for 1 s to rapidly transfer all beads from the sample cup to the dewar.

**Liquid-state NMR measurements at 9.4 T and room temperature.** To demonstrate the necessity of the thermalization step prior to the extraction of polarized solid beads, a series of *ex situ* dissolution experiments were performed. The DNP polarizer was operated at both 4.2 K (Supplementary Fig. 5) and 1 K (Fig. 4). For these experiments, the quartz dewar used to collect the polarized frozen beads was substituted by a glass vial filled with 3 ml of hot $D_2O$, preheated to $100\,°C$ and placed at the centre of a 100 mT permanent magnet. After extraction, the frozen beads immediately melted in contact of the hot $D_2O$. The glass vial and the permanent magnet were then transported 10 m to be placed in the vicinity of a horizontal-bore actively shielded 9.4 T rodent MRI scanner (Varian, Palo Alto, California). The hyperpolarized $^{13}C$ solution was extracted from the vial and injected into a custom-made injection pump equipped with a $^{13}C$ NMR probe (Alderman-Grant coil tuned to 100.67 MHz)[27,37]. The delay between the extraction of the solid sample and the injection of the solution was 15 s. The hyperpolarized $^{13}C$ signal evolution was monitored by applying a 15° rf pulse every 3 s. After a time interval sufficient for a complete relaxation of the $^{13}C$ spins (set to 10 min), the thermally polarized NMR signal was measured using the same 15° pulse (100 averages with a repetition time of 300 s, which is equivalent to about $5 \cdot T_1$). The DNP enhancement was obtained by computing the ratio between the signal integral of the first measured spectrum and the thermally polarized spectrum. Several control experiments ($n = 4$) confirmed that no $^{13}C$ signal could be detected after a single 15° pulse if the rapid thermalization step was omitted from the whole procedure.

**Data availability.** All data are available from the authors upon reasonable request.

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

## Acknowledgements

We thank Professor Jean-Noel Hyacinthe and Dr Davide Antonio Cucci for their useful advices with MATLAB. This work is part of a project that has received funding from the European Union's Horizon 2020 European Research Council (ERC Consolidator Grant) under grant agreement No 682574 (ASSIMILES), the Swiss National Science Foundation (grant PP00P2_133562) and the Centre d'Imagerie BioMédicale (CIBM) of the UNIL, UNIGE, HUG, CHUV, EPFL, and the Leenards and Jeantet Foundations.

## Author contributions

A. Capozzi prepared the ultraviolet-irradiated samples, built the hardware to perform the thermalization and extraction processes. A. Capozzi and T.C. performed the solid-state DNP and ESR measurements to optimize the radical annihilation process. A. Capozzi and T.C. optimized the solid sample extraction and *ex situ* dissolution procedure and performed the *in vitro* $^{13}C$ hyperpolarized MRS measurements. A. Capozzi and G.B. designed and performed the ESR experiments to determine the radical annihilation temperature. A. Capozzi, C.R. and A. Comment designed the study. A. Capozzi and A. Comment analysed the data and wrote the manuscript.

## Additional information

**Competing interests:** A. Comment is currently employed by General Electric Medical Systems, Inc. The remaining authors declare no competing financial interests.

**Publisher's note**: 

