## [Peer Review File · Nature Communications]

Reviewers' comments:

Reviewer #1 (Remarks to the Author):

The manuscript by Comment et al. refers about an advancement in the field of DNP hyperpolarization of pyruvate without the use of exogenous radicals, a method that has been recently reported and relies on irradiation of frozen pyruvate with UV, to obtain photo-induced radicals. At the end of the hyperpolarization, the radicals are recombined during dissolution.

In the herein presented manuscript, an intermediate step named thermalization is introduced, that allows quenching of the radicals while keeping the sample in the solid state. This allows to considerably extend the hyperpolarization lifetime and the ^{13}C T_1 is increased to more than 16h. Hence, remote hyperpolarization of pyruvate and long distance transportation of the HP metabolite is made possible. I think these results deserve publication in Nat. Comm.

This is a relevant advancement in the field of application of hyperpolarized metabolites, nevertheless some weakness points, intrinsic to the polarization without exogenous radicals, can be evidenced:

- long polarization time (3h) is necessary to reach the maximum polarization, furthermore, the time necessary for the whole hyperpolarization process is further increased by the UV irradiation time (60min).

- The maximum polarization obtained at 1.4K is "only" 12% (8.3 after dissolution). This is less than half of the ^{13}C hyperpolarization usually reported for pyruvate obtained using exogenous radicals.

Maybe the ^{13}C hyperpolarization level and the polarization time can be improved by ^1H - ^{13}C cross-polarization, as reported by Bornet et al. (J. Phys. Chem. Lett. 2013, 4, 111–114)? Some more comments may be added on these items.

Reviewer #2 (Remarks to the Author):

This is a very good contribution in the field of Dynamic Nuclear Polarization (DNP) demonstrating that the lifetime of ^{13}C hyperpolarized agent can be significantly extended to over many hours, and can exceed more than 10 hours. This would be interesting to many working in this field. I am very much in favor of publishing this contribution. If this manuscript had submitted to JPC, I would recommend its acceptance minus the "insults" to the field of molecular imaging dominated by the PET community rather than ^{13}C DNP and expanding the references section. Given that this is Nature Communications, I am recommending a minor revision to additionally voice my criticism that the manuscript's introduction should be re-written in the broader context of hyperpolarization rather than dissolution DNP alone. The findings and the work conducted though is of the highest caliber and the manuscript / this contribution will be of broad interest to scientists around the world. I note that more than 100 DNP systems have been sold around the globe (and many home built setups were developed), so this work would be of immediate relevant to a wide range of scientists: from physicists to medical doctors.

Key Points:

- 1) First sentence of the Main text "Because hyperpolarized ^{13}C MRI is currently the only modality allowing real-time metabolic imaging in vivo": has no merit. Hyperpolarized ^{13}C MRI is the recent metabolic imaging modality, but certainly not "the only". This is a huge insult to the entire field of

molecular imaging, and no credible scientists would be able to accept this sentence. I can name a few techniques: detection of redox and other reactions with optical imaging, EPR imaging of oxygenation states, FMISO PET, hyperpolarized ^{129}Xe sensing of receptor imaging... There is a long list.

2) There is a second issue with this first sentence. ^{13}C is not the only hyperpolarized nucleus that can be used to track metabolism in vivo. I would rather see the authors to articulate, why ^{13}C is so ubiquitous for biomedical uses: i.e. huge chemical shift dispersion, ubiquitous presence in the vast majority of biomolecules and metabolites, and low natural abundance background. This is Nature Communications, and the authors owe it to the reader to guide it to the importance why this work is a big deal (and it is).

3) The literature and the Introduction is narrowly focused on selected eighteen references, a significant fraction of those are self-references. The cited literature is certainly relevant, but it should be expanded by others references. In the context of Ref. #5, please cite other relevant reviews in the field: (i) Kurhanewicz, J.; Vigneron, D. B.; Brindle, K.; Chekmenev, E. Y.; Comment, A.; et al. Analysis of Cancer Metabolism by Imaging Hyperpolarized Nuclei: Prospects for Translation to Clinical Research Neoplasia 2011, 13, 81–97. (ii) Brindle, K. M. Imaging Metabolism with Hyperpolarized ^{13}C -Labeled Cell Substrates. J. Am. Chem. Soc. 2015, 137, 6418–6427. (iii) Nikolaou, P.; Goodson, B. M.; Chekmenev, E. Y. Nmr Hyperpolarization Techniques for Biomedicine. Chem. Eur. J. 2015, 21, 3156–3166. (iv) Witte, C.; Schroder, L. Nmr of Hyperpolarised Probes. NMR Biomed. 2013, 26, 788–802. And potentially others especially those inclusive of other hyperpolarization techniques related to metabolic imaging.

4) When discussing the technical challenge “The fact that the ^{13}C spins are extracted in a liquid form has however some disadvantages: it cannot be transported very far from its production site because of the relatively short relaxation time, typically shorter than a minute;” Please, at least state that there is a option that LLSS were shown (but arguably not on metabolites yet): Warren, W. S.; Jenista, E.; Branca, R. T.; Chen, X. Increasing Hyperpolarized Spin Lifetimes through True Singlet Eigenstates. Science 2009, 323, 1711–1714. At the very least, the manuscript should also discuss that the presented approach is much better than LLSS use. This needs to be discussed, b/c Nature Communications is a top journal, and the authors have to discuss this work in the broad context (certainly within the field of hyperpolarization).

5) Dissolution DNP has certainly been in the center, but not from the very beginning of ^{13}C in vivo HP MRI: Golman, K.; Axelsson, O.; Johannesson, H.; Mansson, S.; Olofsson, C.; Petersson, J. S. Parahydrogen-Induced Polarization in Imaging: Subsecond C-13 Angiography. Magn. Reson. Med. 2001, 46, 1–5. This work should be cited as the pioneering contribution in the field of in vivo ^{13}C MRI.

6) The sentence: “There are two ways of increasing the strength of the emf: by increasing either the precession frequency of the magnetic moments or the amplitude of the nuclear spin magnetization.” is semi-correct and tangential. Why the precession frequency and EMF are so important? In fact, more SNR can be obtained at lower frequencies when using hyperpolarization: Coffey, A. M.; Truong, M. L.; Chekmenev, E. Y. Low-Field Mri Can Be More Sensitive Than High-Field Mri. J. Magn. Reson. 2013, 237, 169–174. One should also remember that SNR is a true value in NMR/MRI rather than EMF itself. Since this paper is focused on DNP rather than sensitivity of detection I would suggest re-wording the sentence to: “The strength of the emf can be increased through the increase of the amplitude of the nuclear spin magnetization.” or something similar. Reword the following sentence accordingly; Note that polarization also increases with the field strength.

7) There are certainly other biomolecules that can be hyperpolarized with DNP. This study is focused on pyruvate. The authors need to comment on the applicability of the demonstrated technique to other biomolecules. This is not done.

8) In part, the reason why authors demonstrate such long T1 in the solid is due to the use of low temperature. This has been studied extensively in the context of ^{129}Xe , and this needs to be discussed: e.g. more recently by Saam and co-workers <https://arxiv.org/pdf/1607.01072.pdf>

Reviewer #3 (Remarks to the Author):

Dr. Andrey N. Pravdivtsev
pravdivtsev.a.n@gmail.com

Review

It is well written paper and a great contribution to the DNP field. It will be of interest to wider audience because it has interesting physical and chemical background and has direct medical application. The paper has well written separate experimental section that will able one to reproduce the experiments. I strongly recommend it for publication in Nature Communication after minor revision.

1. I strongly recommend to use level of polarization (%) instead of enhancement factor because it is absolute unity. It should be done in line 67 and in figure S3. You may specify enhancement in addition if you like.
2. (Lines 75-77). It is unclear from the very beginning that you are describing your own observations but not continuing the introduction. Also write this sentence in clearer way.
3. (Lines 129-132) The purpose is unclear at the very beginning. It is better to simplify the sentence and add one more sentence to clarify why you had to do this experiment and how it will be used in future.
4. Why you obtain polarization level less than unity? At the intro it was pointed out that in 70th polarization level around unity was reached! Could you comment it or suggest an improvement?
5. You made your beads from H₂O. But finally you dissolve the beads in hot D₂O. I think it will be very valuable for the contribution to compare results for beads made from H₂O and D₂O. You have excellent methodology and it should be easy to implement.
6. (line 200) Power of MW-source is 55 mW. Please specify the amplitude in "Hz" that is very important to see which part of spectrum could be covered. You should add ESR spectrum of radical (maybe even calculated one) on the top of the figure S3, when it will be easy to see the part of spectrum that you hit during preparation of DNP.
7. Figure S3. You should add specifications of MW-irradiation: time, power, amplitude (in GHz) and carrier frequency.
8. (line 204 and 206) It is unclear why do you use 30-degree RF-pulse.
9. Figure S4. you did not fully specified MW-irradiation, and there is no information about angle of RF-pulse that you determined here. Section "Solid-state ¹³C NMR coil calibration and rf correction" should be checked: (1) "(see Fig. S3)" should be replaced by "(see Fig. S4)". (2). Text should be before the figure.
10. In the SM H₂O in several places is H₂"Zero".

In addition to that Figures should be revised with care:

1. Font size in figure 2 and 4 seems to be tiny.
2. In all figures with spectra or dependences the temperature should be specified on the figure explicitly.
3. The color of you beads should be the same as the color of corresponding dependences.
i.e.
 - 3.1. Figure 2 (d) cyan as you called it "blueish" color should be replaced by the same blue color like in part (e) and (f), like you do it for "yellowish" beads and corresponding plots. However changes should be consistent with Figure 2-4 which should be modified respectfully.
 - 3.2. Figure 4 (b) Probably here color should be yellow? Because it was measured before thermalization.
 - 3.3. Figure 4 (f) color should be blue because it is after thermalization.

etc for all Figures.

Now in the text you mix up "cold" and "warm" colors and "before" and "after" thermalization. It is better to use one color coding.

Despite the above mentioned criticism

it was pleasure to review the manuscript.

Reviewer #4 (Remarks to the Author):

The authors have cleverly and decisively demonstrated for the first time that it is possible to extract a frozen sample of pyruvic acid (PA) from the polarizer before performing a solid-to-liquid phase transition. The authors have produced transient photo-induced radicals for the polarization of PA which can be annihilated by raising the sample temperature while it is in a solid state. This process—hyperpolarizing a PA sample at 1K with transient radicals produced through UV-irradiation—is a dramatic advancement in the dynamic nuclear polarization (DNP) of PA for the real-time study of metabolic pathways *in vivo*, and has many obvious advantages.

A method for hyperpolarizing carbon-13 compounds through DNP based on the use of non-persistent radicals is in many ways an ideal one. The current method for polarizing carbon-13 compounds using DNP is to mix the target molecule with a radical and then radiate the frozen sample with microwave frequency at a temperature of $\sim 1\text{K}$. After achieving the desired level of polarization, the sample is then extracted for *in vitro* or *in vivo* studies after *in situ* dissolution. While this method has been used effectively for studying metabolic pathways *in vivo* in real-time, it has several disadvantages. For one, *in situ* dissolution is a very complex process that complicates the *in-vivo* applications of DNP studies. For example, a minor mistake in following the complex dissolution protocol leads to frozen lines that cause system downtime, failed dissolution and unsuccessful studies. This is an extremely important factor for human studies, where subject preparation is highly complex and the availability of a robust dissolution method is critical. Second, *in situ* dissolution leads to the extraction of the polarized sample in liquid form, limiting its usage due to the very short T_1 relaxation time of hyperpolarized PA in this state. Finally, any clinical applications require that the radicals be completely separated from the sample prior its administration, a process which could lead to a loss of polarization. This paper effectively addresses all of these limitations.

The paper displays a sound research design, from which the authors have derived very exciting results. Using their established method of polarizing PA with non-persistent radicals produced by UV radiation of the sample, they have shown that this method can be modified to annihilate the radicals by changing the sample temperature. Figure 1 powerfully demonstrates the presence and absence of radicals at temperatures below and above 192K, respectively. This is an ideal method for polarizing PA for *in-vivo* studies.

Due to the significant difference in T_1 relaxation time between the liquid and solid phases of hyperpolarized PA (tens of seconds in the liquid phase, as opposed to many tens of minutes in the solid state), this method could dramatically expand the use of this technology for medical and non-medical applications; taking advantage of the much longer T_1 relaxation time of solid hyperpolarized PA, it allows a frozen sample to be extracted from the polarizer and transferred to another location without a significant loss of polarization. This is a major advantage, since the polarizer and the MRI scanner no longer need to be in close physical proximity to one another. Indeed, this method means that it is now possible to use one polarizer to produce polarized samples for use in multiple scanners in multiple locations.

While the paper effectively demonstrates the capabilities of the described method, it does not do as effective a job of discussing its limitations, which should be briefly discussed in the manuscript. These limitations include:

1. The low polarization percentage currently achievable with this method. A look at the X-band ESR spectrum of the PA sample collected at 77 K indicates that this method will never be able to produce the same polarization level that can be achieved when using a trityl radical. This represents a fundamental limitation of the described method; despite the fact that they have used a 7 T DNP system, the somewhat unfavorable level of polarization they report (around 12%) could dramatically limit this method's clinical applications. It might be useful to discuss the attributes of the ESR spectra provided here compared to the ESR spectra of a trityl radical. Specifically, how would these changes in ESR spectra affect the polarization?

2. The method demonstrated here is currently only applicable to a finite number of molecules, so

the authors should discuss why it cannot be used for other interesting biological molecules. In fact, one could argue that molecules with low solubility may benefit from this method more significantly, as it does not require a uniform mixture of radical and carbon-13 compounds.

In summary, this is an outstanding piece of work. Though the above suggestions would make it more accessible to general readers, this is an excellent manuscript even in its present state.

Reviewers' comments:

Reviewer #1 (Remarks to the Author):

The manuscript by Comment et al. refers about an advancement in the field of DNP hyperpolarization of pyruvate without the use of exogenous radicals, a method that has been recently reported and relies on irradiation of frozen pyruvate with UV, to obtain photo-induced radicals. At the end of the hyperpolarization, the radicals are recombined during dissolution.

In the herein presented manuscript, an intermediate step named thermalization is introduced, that allows quenching of the radicals while keeping the sample in the solid state. This allows to considerably extend the hyperpolarization lifetime and the ^{13}C T_1 is increased to more than 16h. Hence, remote hyperpolarization of pyruvate and long distance transportation of the HP metabolite is made possible. I think these results deserve publication in Nat. Comm.

This is a relevant advancement in the field of application of hyperpolarized metabolites, nevertheless some weakness points, intrinsic to the polarization without exogenous radicals, can be evidenced:

- long polarization time (3h) is necessary to reach the maximum polarization, furthermore, the time necessary for the whole hyperpolarization process is further increased by the UV irradiation time (60min).**
- The maximum polarization obtained at 1.4K is "only" 12% (8.3 after dissolution). This is less than half of the ^{13}C hyperpolarization usually reported for pyruvate obtained using exogenous radicals.**

Maybe the ^{13}C hyperpolarization level and the polarization time can be improved by ^1H - ^{13}C cross-polarization, as reported by Bornet et al. (J. Phys. Chem. Lett. 2013, 4, 111–114)? Some more comments may be added on these items.

We would first like to thank the reviewer for his valuable comments. We agree that the lower polarization level and longer polarization time are currently two important limitations of the

proposed method. However, because the samples can be irradiated days in advance and stored in liquid nitrogen, we do not think that the time required to perform the UV irradiation should be included in the calculation of the effective time for the DNP process. We would also like to emphasize that the irradiation time is directly linked to the radical quantum yield, which will depend on the light source used for the irradiation. It is therefore not yet clear what the minimum irradiation time would be for an optimal UV source, but a complete optical study is beyond the scope of this manuscript.

We have added the following text before the concluding paragraph to address these limitations:

“The liquid-state ^{13}C polarization obtained in the present work is severalfold lower than the maximum values obtained by dissolution DNP using the same polarizer and trityl radicals (Yoshihara, Can et al. 2016). The narrower ESR line width of trityl radicals as compared to UV-induced radicals is expected to lead to higher ^{13}C polarization (Eichhorn, Takado et al. 2013), but the main reason for the lower polarization is the sub-optimal radical concentration, which also explains that the build-up time constant is larger than in an optimized PA sample prepared with trityl radical. Preliminary results show that this concentration could be improved by using a broadband UV source, but a complete optical study is beyond the scope of this manuscript. Modulating the microwave frequency or performing ^1H - ^{13}C cross-polarization could also improve the polarization level and reduce the build-up time (Bornet, Melzi et al. 2013, Bornet, Milani et al. 2014), but these methods require additional hardware.”

Reviewer #2 (Remarks to the Author):

This is a very good contribution in the field of Dynamic Nuclear Polarization (DNP) demonstrating that the lifetime of ^{13}C hyperpolarized agent can be significantly extended to over many hours, and can exceed more than 10 hours. This would be interesting to many working in this field. I am very much in favor of publishing this contribution. If this manuscript had submitted to JPC, I would recommend its acceptance minus the “insults” to the field of molecular imaging dominated by the PET community rather than ^{13}C DNP and expanding the references section. Given that this is

Nature Communications, I am recommending a minor revision to additionally voice my criticism that the manuscript's introduction should be re-written in the broader context of hyperpolarization rather than dissolution DNP alone. The findings and the work conducted though is of the highest caliber and the manuscript / this contribution will be of broad interest to scientists around the world. I note that more than 100 DNP systems have been sold around the globe (and many home built setups were developed), so this work would be of immediate relevant to a wide range of scientists: from physicists to medical doctors.

Key Points:

1) First sentence of the Main text "Because hyperpolarized ^{13}C MRI is currently the only modality allowing real-time metabolic imaging in vivo": has no merit. Hyperpolarized ^{13}C MRI is the recent metabolic imaging modality, but certainly not "the only". This is a huge insult to the entire field of molecular imaging, and no credible scientists would be able to accept this sentence. I can name a few techniques: detection of redox and other reactions with optical imaging, EPR imaging of oxygenation states, FMISO PET, hyperpolarized ^{129}Xe sensing of receptor imaging... There is a long list.

We agree with the reviewer that the introduction should be expanded to cover all molecular imaging modalities, even those that have not yet been used clinically. We have revised the manuscript accordingly. We have also incorporated the references mentioned in the reviewer's point #4, included alternative hyperpolarization techniques (to take into account reviewer's point #5) and emphasized the reasons why ^{13}C is the most versatile nucleus for biomedical applications (reviewer's point #2). The first part of the introduction has therefore been replaced by the following:

"Hyperpolarized ^{13}C MRI is one among several molecular imaging techniques proposed in the recent years to detect biochemical changes *in vivo* (Golman, in't Zandt et al. 2006, Nelson, Kurhanewicz et al. 2013). Most imaging modalities, including MRI, computed tomography (CT), positron emission tomography (PET), single photon emission computed tomography (SPECT), ultrasound, and a variety of optical imaging methods can be adapted to reveal insights into cellular function (James and Gambhir 2012). MRI can provide direct biochemical information via the spectroscopic dimension of nuclear magnetic resonance (NMR), allowing simultaneous acquisition

of signals from a substrate and its metabolic products, hence yielding true metabolic imaging. The relatively low sensitivity of NMR can be circumvented using hyperpolarization techniques such as spin-exchange optical pumping (SEOP) for gases, and dissolution DNP, parahydrogen-induced polarization (PHIP) as well as the so-called "brute force" method for liquids (Comment 2013, Witte and Schroder 2013, Nikolaou, Goodson et al. 2015). In the context of metabolic imaging, ^{13}C is the most adapted nucleus because of its ubiquitous presence in the vast majority of biomolecules, its large chemical shift dispersion allowing to easily differentiate the various species, its low natural abundance, and its relatively long longitudinal relaxation time at specific molecular positions such as in a carboxyl group (Golman, Olsson et al. 2003, Golman, in't Zandt et al. 2006, Mansson, Johansson et al. 2006, Kurhanewicz, Vigneron et al. 2011, Comment and Merritt 2014, Brindle 2015). PHIP was the first hyperpolarization technique proposed for *in vivo* ^{13}C MRI (Golman, Axelsson et al. 2001), but dissolution DNP became more popular because of its versatility for applications in biomedical imaging (Ardenkjaer-Larsen, Fridlund et al. 2003, Ardenkjaer-Larsen 2016)."

2) There is a second issue with this first sentence. ^{13}C is not the only hyperpolarized nucleus that can be used to track metabolism *in vivo*. I would rather see the authors to articulate, why ^{13}C is so ubiquitous for biomedical uses: i.e. huge chemical shift dispersion, ubiquitous presence in the vast majority of biomolecules and metabolites, and low natural abundance background. This is Nature Communications, and the authors owe it to the reader to guide it to the importance why this work is a big deal (and it is).

See answer to point #1.

3) The literature and the Introduction is narrowly focused on selected eighteen references, a significant fraction of those are self-references. The cited literature is certainly relevant, but it should be expanded by others references. In the context of Ref. #5, please cite other relevant reviews in the field: (i) Kurhanewicz, J.; Vigneron, D. B.; Brindle, K.; Chekmenev, E. Y.; Comment, A.; et al. Analysis of Cancer Metabolism by Imaging Hyperpolarized Nuclei: Prospects for Translation to Clinical Research Neoplasia 2011, 13, 81–97. (ii) Brindle, K. M. Imaging Metabolism with Hyperpolarized ^{13}C -Labeled Cell Substrates. J. Am. Chem. Soc. 2015, 137, 6418–6427. (iii) Nikolaou, P.; Goodson, B. M.; Chekmenev, E. Y. Nmr Hyperpolarization Techniques for Biomedicine.

Chem. Eur. J. 2015, 21, 3156-3166. (iv) Witte, C.; Schroder, L. Nmr of Hyperpolarised Probes. NMR Biomed. 2013, 26, 788-802. And potentially others especially those inclusive of other hyperpolarization techniques related to metabolic imaging.

See answer to point #1.

4) When discussing the technical challenge "The fact that the ^{13}C spins are extracted in a liquid form has however some disadvantages: it cannot be transported very far from its production site because of the relatively short relaxation time, typically shorter than a minute;" Please, at least state that there is a option that LLSS were shown (but arguably not on metabolites yet): Warren, W. S.; Jenista, E.; Branca, R. T.; Chen, X. Increasing Hyperpolarized Spin Lifetimes through True Singlet Eigenstates. Science 2009, 323, 1711-1714. At the very least, the manuscript should also discuss that the presented approach is much better than LLSS use. This needs to be discussed, b/c Nature Communications is a top journal, and the authors have to discuss this work in the broad context (certainly within the field of hyperpolarization).

We added the following note after the sentence mentioned by the reviewer:

"...(this time can be extended in some specific molecules through the formation of a so-called long-lived state(Vasos, Comment et al. 2009, Warren, Jenista et al. 2009), but no decisive application for *in vivo* metabolic imaging has been proposed to date(Marco-Rius, Tayler et al. 2013));"

5) Dissolution DNP has certainly been in the center, but not from the very beginning of ^{13}C in vivo HP MRI: Golman, K.; Axelsson, O.; Johannesson, H.; Mansson, S.; Olofsson, C.; Petersson, J. S. Parahydrogen-Induced Polarization in Imaging: Subsecond C-13 Angiography. Magn. Reson. Med. 2001, 46, 1-5. This work should be cited as the pioneering contribution in the field of in vivo ^{13}C MRI.

We agree with the reviewer and have included these references (see our answer to point #1).

6) The sentence: "There are two ways of increasing the strength of the emf: by increasing either the precession frequency of the magnetic moments or the amplitude of the nuclear spin magnetization." is semi-correct and tangential. Why the precession frequency and EMF are so important? In fact, more SNR can be obtained at lower

frequencies when using hyperpolarization: Coffey, A. M.; Truong, M. L.; Chekmenev, E. Y. Low-Field Mri Can Be More Sensitive Than High-Field Mri. J. Magn. Reson. 2013, 237, 169-174. One should also remember that SNR is a true value in NMR/MRI rather than EMF itself. Since this paper is focused on DNP rather than sensitivity of detection I would suggest re-wording the sentence to: "The strength of the emf can be increased through the increase of the amplitude of the nuclear spin magnetization." or something similar. Reword the following sentence accordingly; Note that polarization also increases with the field strength.

We replaced the sentence by one similar to the sentence suggested by the reviewer, i.e.: "The strength of the emf can be enhanced by increasing the amplitude of the nuclear spin polarization,..."

7) There are certainly other biomolecules that can be hyperpolarized with DNP. This study is focused on pyruvate. The authors need to comment on the applicability of the demonstrated technique to other biomolecules. This is not done.

We are currently working on extending this method to other compounds beyond pyruvic acid and we already know that most alpha-keto acids should have similar properties after UV irradiation, f.i. 2-keto isocaproic acid (Steffen F. Frank et al., Hyperpolarization of 2-keto[1-¹³C]isocaproate for in vivo studies with photo-induced radicals, abstract accepted for presentation at the upcoming ISMRM 25th Annual meeting) or oxaloacetic acid, as well as some other alpha-keto acids that are not metabolized by mammalian organism and could be mixed with other molecules of interest in a fashion similar to what was demonstrated by Capozzi et al. (J. Phys. Chem. C 119, 22632, 2015) with pyruvic acid. These studies are still ongoing and will be presented in future publications. We have however added the following sentence in the conclusion:

"The herein presented technique can be used to polarize other ¹³C-compounds as well as other nuclei following the method proposed by Capozzi *et al.* (Capozzi, Hyacinthe et al. 2015). Since other alpha-keto acids exhibit similar photochemical behavior as PA (Leermakers and Vesley 1963), we expect to be able to extend this technique to other biomolecules in the near future."

8) In part, the reason why authors demonstrate such long T1 in the solid is due to the use of low temperature. This has been studied extensively in the context of ¹²⁹Xe, and

this needs to be discussed: e.g. more recently by Saam and co-workers

<https://arxiv.org/pdf/1607.01072.pdf>

We have added a sentence to highlight the similarity with hyperpolarized ^{129}Xe :

“Note that it has also been demonstrated that hyperpolarized ^{129}Xe can be stored for an extended period of time and transported in the solid form at cryogenic temperature (Hersman, Ruset et al. 2008, Limes, Ma et al. 2016).”

Reviewer #3 (Remarks to the Author):

Dr. Andrey N. Pravdivtsev

pravdivtsev.a.n@gmail.com

Review

It is well written paper and a great contribution to the DNP field. It will be of interest to wider audience because it has interesting physical and chemical background and has direct medical application. The paper has well written separate experimental section that will able one to reproduce the experiments. I strongly recommend it for publication in Nature Communication after minor revision.

1. I strongly recommend to use level of polarization (%) instead of enhancement factor because it is absolute unity. It should be done in line 67 and in figureS3. You may specify enhancement in addition if you like.

We fully agree with the reviewer that polarization level should be used because it is an absolute value. Supplementary Figure 3 has been modified accordingly. All other results presented in our manuscript are expressed in % polarization. In line 67, where we refer to work published by Ardenkjaer-Larsen *et al.* (PNAS 2003), we think that it would be incorrect to specify a polarization level instead of the enhancement factor the authors chose to report in their original manuscript.

2. (Lines 75-77). It is unclear from the very beginning that you are describing your own observations but not continuing the introduction. Also write this sentence in clearer way.

We have replaced the sentence by the following two sentences:

“These radicals, formed at 77 K by UV-irradiation of frozen aliquots containing PA, do not persist when brought to room temperature. Using electron spin resonance (ESR), we determined that the radicals are annihilated if the sample temperature increases above 190 ± 2 K (Fig. 1).”

3. (Lines 129-132) The purpose is unclear at the very beginning. It is better to simplify the sentence and add one more sentence to clarify why you had to do this experiment and how it will be used in future.

We have replaced the ambiguous sentence by the following two sentences:

“The elimination of the radicals in the solid state alleviates the crucial requirement for dissolution to take place inside the high-field and low-temperature environment of the polarizer. To demonstrate that it is indeed possible to perform the solid-to-liquid transformation outside of the polarizer after extraction of the frozen beads without considerable reduction of the ^{13}C polarization, an additional set of experiments was performed.”

4. Why you obtain polarization level less than unity? At the intro it was pointed out that in 70th polarization level around unity was reached! Could you comment it or suggest an improvement?

We have added the following text before the concluding paragraph to address these limitations and discuss potential improvements:

“The liquid-state ^{13}C polarization obtained in the present work is severalfold lower than the maximum values obtained by dissolution DNP using the same polarizer and trityl radicals (Yoshihara, Can et al. 2016). The narrower ESR line width of trityl radicals as compared to UV-induced radicals is expected to lead to higher ^{13}C polarization (Eichhorn, Takado et al. 2013), but the main reason for the lower polarization is the sub-optimal radical concentration, which also explains that the build-up time constant is larger than in an optimized PA sample prepared with trityl radical. Preliminary results show that this concentration could be improved by using a broadband UV source, but a complete optical study is beyond the scope of this manuscript. Modulating the microwave frequency or performing ^1H - ^{13}C cross-polarization could also improve

the polarization level and reduce the build-up time (Bornet, Melzi et al. 2013, Bornet, Milani et al. 2014), but these methods require additional hardware."

5. You made your beads from H₂O. But finally you dissolve the beads in hot D₂O. I think it will be very valuable for the contribution to compare results for beads made from H₂O and D₂O. You have excellent methodology and it should be easy to implement.

We agree with the reviewer that replacing H₂O by D₂O is indeed an interesting experiment. In fact, we had already performed such experiment but did not include the results in the manuscript. The reason is that there was no statistically significant difference between the two types of samples. Perhaps the results would be more conclusive if PA was also replaced by deuterated PA, but we think that this would not be of great interest for future in vivo studies and it would be beyond the scope of this manuscript. We nevertheless added the following sentence in the Methods section:

"Note that the observed enhancement and time evolution were similar for samples composed of [1-¹³C]PA:D₂O 1:1 (v/v) and we therefore only focused on samples prepared with H₂O."

6. (line 200) Power of MW-source is 55 mW. Please specify the amplitude in "Hz" that is very important to see which part of spectrum could be covered. You should add ESR spectrum of radical (maybe even calculated one) on the top of the figure S3, when it will be easy to see the part of spectrum that you hit during preparation of DNP.

The microwave B₁ field is ill-defined in this type of DNP experiments since the sample is larger than the wavelength and there is no resonant cavity to perform ESR experiments. We however agree with the reviewer that adding the calculated ESR spectrum in Supplementary Figure 3 makes it easy to compare the microwave spectrum with the width of the ESR line width. The figure has been updated accordingly.

7. Figure S3. You should add specifications of MW-irradiation: time, power, amplitude (in GHz) and carrier frequency.

The figure legend has been modified accordingly:

"DNP microwave spectrum. ¹³C DNP microwave spectrum (open circles connected by black segments) measured at 7 T and 4.2 K in UV-irradiated [1-¹³C]PA:H₂O 1:1 (v/v). The ¹³C polarization was determined for each microwave frequency after 60 min of irradiation (microwave output power set to 55 mW) using the method described in the main text (see Methods). The grey curve represents the ESR spectrum of the photo-induced radical calculated at 7 T with the PEPPER routine of the MATLAB®-based software EasySpin2 using the g-tensor ([2.0041 2.0037 2.0042]), ¹H hyperfine coupling (48 MHz isotropic to each of the 3 methyl group protons), ¹³C hyperfine coupling (30 MHz isotropic), and Gaussian line broadening (20 MHz) values obtained from X-band ESR measurements (see Supporting Information in ref. 2, DOI: 10.1021/acs.jpcc.5b07315)."

8. (line 204 and 206) It is unclear why do you use 30-degree RF-pulse.

A larger flip angle is necessary to increase the SNR of the thermal equilibrium signal in order to accurately determine the solid-state polarization. We added the following sentence to clarify this point:

"The DNP enhancement was obtained by computing the ratio between the two NMR signal integrals. A larger flip angle (30°) was necessary to increase the signal-to-noise ratio of the thermal equilibrium signal in order to accurately determine the solid-state polarization."

9. Figure S4. you did not fully specified MW-irradiation, and there is no information about angle of RF-pulse that you determined here. Section "Solid-state ¹³C NMR coil calibration and rf correction" should be checked: (1) "(see Fig. S3)" should be replaced by "(see Fig. S4)". (2). Text should be before the figure.

We thank the reviewer for noticing these errors and missing information. "see Fig. S3" was replaced by "see Fig. S4" and the text moved before Fig. S4. The legend was replaced by the following:

"¹³C NMR signal integral as a function of the acquisition number. The consecutive single-scan acquisitions spaced by 1 s. rf excitation was done with a single 5 μs and 6.3 W square pulse, corresponding to a flip angle of 6.0±0.5°. The measurements were performed at 4.2 K and 7 T after having partially polarized the sample for 10 min with 55 mW microwave power (as measured

at the source output) at 196.633 GHz (optimal frequency according to Supplementary Fig. 3)."

10. In the SM H₂O in several places is H₂"Zero".

In addition to that Figures should be revised with care:

1. Font size in figure 2 and 4 seems to be tiny.

2. In all figures with spectra or dependences the temperature should be specified on the figure explicitly.

3. The color of you beads should be the same as the color of corresponding dependences.

i.e.

3.1. Figure 2 (d) cyan as you called it "blueish" color should be replaced by the same blue color like in part (e) and (f), like you do it for "yellowish" beads and corresponding plots. However changes should be consistent with Figure 2-4 which should be modified respectfully.

3.2. Figure 4 (b) Probably here color should be yellow? Because it was measured before thermalization.

3.3. Figure 4 (f) color should be blue because it is after thermalization.

etc for all Figures.

Now in the text you mix up "cold" and "warm" colors and "before" and "after" thermalization. It is better to use one color coding.

We updated the figures and manuscript according to the reviewer's suggestions.

Despite the above mentioned criticism

it was pleasure to review the manuscript.

We thank the reviewer for his positive comment.

Reviewer #4 (Remarks to the Author):

The authors have cleverly and decisively demonstrated for the first time that it is possible to extract a frozen sample of pyruvic acid (PA) from the polarizer before

performing a solid-to-liquid phase transition. The authors have produced transient photo-induced radicals for the polarization of PA which can be annihilated by raising the sample temperature while it is in a solid state. This process—hyperpolarizing a PA sample at 1K with transient radicals produced through UV-irradiation—is a dramatic advancement in the dynamic nuclear polarization (DNP) of PA for the real-time study of metabolic pathways in vivo, and has many obvious advantages.

A method for hyperpolarizing carbon-13 compounds through DNP based on the use of non-persistent radicals is in many ways an ideal one. The current method for polarizing carbon-13 compounds using DNP is to mix the target molecule with a radical and then radiate the frozen sample with microwave frequency at a temperature of $\sim 1\text{K}$. After achieving the desired level of polarization, the sample is then extracted for in vitro or in vivo studies after in situ dissolution. While this method has been used effectively for studying metabolic pathways in vivo in real-time, it has several disadvantages. For one, in situ dissolution is a very complex process that complicates the in-vivo applications of DNP studies. For example, a minor mistake in following the complex dissolution protocol leads to frozen lines that cause system downtime, failed dissolution and unsuccessful studies. This is an extremely important factor for human studies, where subject preparation is highly complex and the availability of a robust dissolution method is critical. Second, in situ dissolution leads to the extraction of the polarized sample in liquid form, limiting its usage due to the very short T1 relaxation time of hyperpolarized PA in this state. Finally, any clinical applications require that the radicals be completely separated from the sample prior its administration, a process which could lead to a loss of polarization. This paper effectively addresses all of these limitations.

The paper displays a sound research design, from which the authors have derived very exciting results. Using their established method of polarizing PA with non-persistent radicals produced by UV radiation of the sample, they have shown that this method can be modified to annihilate the radicals by changing the sample temperature. Figure 1 powerfully demonstrates the presence and absence of radicals at temperatures below and above 192K, respectively. This is an ideal method for polarizing PA for in-vivo studies.

Due to the significant difference in T1 relaxation time between the liquid and solid phases of hyperpolarized PA (tens of seconds in the liquid phase, as opposed to many

tens of minutes in the solid state), this method could dramatically expand the use of this technology for medical and non-medical applications; taking advantage of the much longer T1 relaxation time of solid hyperpolarized PA, it allows a frozen sample to be extracted from the polarizer and transferred to another location without a significant loss of polarization. This is a major advantage, since the polarizer and the MRI scanner no longer need to be in close physical proximity to one another. Indeed, this method means that it is now possible to use one polarizer to produce polarized samples for use in multiple scanners in multiple locations.

While the paper effectively demonstrates the capabilities of the described method, it does not do as effective a job of discussing its limitations, which should be briefly discussed in the manuscript. These limitations include:

1. The low polarization percentage currently achievable with this method. A look at the X-band ESR spectrum of the PA sample collected at 77 K indicates that this method will never be able to produce the same polarization level that can be achieved when using a trityl radical. This represents a fundamental limitation of the described method; despite the fact that they have used a 7 T DNP system, the somewhat unfavorable level of polarization they report (around 12%) could dramatically limit this method's clinical applications. It might be useful to discuss the attributes of the ESR spectra provided here compared to the ESR spectra of a trityl radical. Specifically, how would these changes in ESR spectra affect the polarization?

We first would like to thank the reviewer for his positive feedback.

Regarding his first point, we agree that the discussion should be extended. We have added the following text before the concluding paragraph to discuss the ESR spectra and how the current method could be possibly improved:

"The liquid-state ^{13}C polarization obtained in the present work is severalfold lower than the maximum values obtained by dissolution DNP using the same polarizer and trityl radicals (Yoshihara, Can et al. 2016). The narrower ESR line width of trityl radicals as compared to UV-induced radicals is expected to lead to higher ^{13}C polarization (Eichhorn, Takado et al. 2013), but the main reason for the lower polarization is the sub-optimal radical concentration, which also explains that the build-up time constant is larger than in an optimized PA sample prepared with

trityl radical. Preliminary results show that this concentration could be improved by using a broadband UV source, but a complete optical study is beyond the scope of this manuscript. Modulating the microwave frequency or performing ^1H - ^{13}C cross-polarization could also improve the polarization level and reduce the build-up time (Bornet, Melzi et al. 2013, Bornet, Milani et al. 2014), but these methods require additional hardware."

2. The method demonstrated here is currently only applicable to a finite number of molecules, so the authors should discuss why it cannot be used for other interesting biological molecules. In fact, one could argue that molecules with low solubility may benefit from this method more significantly, as it does not require a uniform mixture of radical and carbon-13 compounds.

We are currently working on extending this method to other compounds beyond pyruvic acid and we already know that most alpha-keto acids should have similar properties after UV irradiation, f.i. 2-keto isocaproic acid (Steffen F. Frank et al., Hyperpolarization of 2-keto[1- ^{13}C]isocaproate for in vivo studies with photo-induced radicals, abstract accepted for presentation at the upcoming ISMRM 25th Annual meeting) or oxaloacetic acid, as well as some other alpha-keto acids that are not metabolized by mammalian organism and could be mixed with other molecules of interest in a fashion similar to what was demonstrated by Capozzi et al. (J. Phys. Chem. C 119, 22632, 2015) with pyruvic acid. These studies are still ongoing and will be presented in future publications.

However, as suggested by the reviewer, we have added the following sentence in the conclusion to discuss the possible extension of the method:

"The herein presented technique can be used to polarize other ^{13}C -compounds as well as other nuclei following the method proposed by Capozzi *et al.* (Capozzi, Hyacinthe et al. 2015). Since other alpha-keto acids exhibit similar photochemical behavior as PA (Leermakers and Vesley 1963), we expect to be able to extend this technique to other biomolecules in the near future."

We also added a sentence to highlight the reviewer's point, which we fully agree with, concerning the potential of this method for molecules with low solubility:

"In this study, we take advantage of the non-persistent nature of specific photo-induced radicals to produce hyperpolarized ^{13}C -substrates that can be extracted from the DNP apparatus without the need for a dissolution process. The frozen solid containing hyperpolarized ^{13}C -substrates can consequently be melted at a later time, in a remote location, and this method also extends the

potential applications of hyperpolarized ^{13}C MRI to biomolecules with low solubility.”

In summary, this is an outstanding piece of work. Though the above suggestions would make it more accessible to general readers, this is an excellent manuscript even in its present state.

We thank the reviewer for his comment.

List of added references (alphabetical order)

- Ardenkjaer-Larsen, J. H. (2016). "On the present and future of dissolution-DNP." J Magn Reson **264**: 3-12.
- Ardenkjaer-Larsen, J. H., B. Fridlund, A. Gram, G. Hansson, L. Hansson, M. H. Lerche, R. Servin, M. Thaning and K. Golman (2003). "Increase in signal-to-noise ratio of > 10,000 times in liquid-state NMR." Proc Natl Acad Sci U S A **100**(18): 10158-10163.
- Bornet, A., R. Melzi, A. J. P. Linde, P. Hautle, B. van den Brandt, S. Jannin and G. Bodenhausen (2013). "Boosting Dissolution Dynamic Nuclear Polarization by Cross Polarization." Journal of Physical Chemistry Letters **4**(1): 111-114.
- Bornet, A., J. Milani, B. Vuichoud, A. J. P. Linde, G. Bodenhausen and S. Jannin (2014). "Microwave frequency modulation to enhance Dissolution Dynamic Nuclear Polarization." Chemical Physics Letters **602**: 63-67.
- Brindle, K. M. (2015). "Imaging Metabolism with Hyperpolarized C-13-Labeled Cell Substrates." Journal of the American Chemical Society **137**(20): 6418-6427.
- Capozzi, A., J. N. Hyacinthe, T. Cheng, T. R. Eichhorn, G. Boero, C. Roussel, J. J. van der Klink and A. Comment (2015). "Photoinduced Nonpersistent Radicals as Polarizing Agents for X-Nuclei Dissolution Dynamic Nuclear Polarization." Journal of Physical Chemistry C **119**(39): 22632-22639.
- Comment, A. (2013). CHAPTER 9 Hyperpolarization: Concepts, Techniques and Applications. New Applications of NMR in Drug Discovery and Development, The Royal Society of Chemistry: 252-272.
- Comment, A. and M. E. Merritt (2014). "Hyperpolarized Magnetic Resonance as a Sensitive Detector of Metabolic Function." Biochemistry **53**(47): 7333-7357.
- Eichhorn, T. R., Y. Takado, N. Salameh, A. Capozzi, T. Cheng, J. N. Hyacinthe, M. Mishkovsky, C. Roussel and A. Comment (2013). "Hyperpolarization without persistent radicals for in vivo real-time metabolic imaging." Proc Natl Acad Sci U S A **110**(45): 18064-18069.
- Golman, K., O. Axelsson, H. Johannesson, S. Mansson, C. Olofsson and J. S. Petersson (2001). "Parahydrogen-induced polarization in imaging: Subsecond C-13 angiography." Magnetic Resonance in Medicine **46**(1): 1-5.
- Golman, K., R. in't Zandt and M. Thaning (2006). "Real-time metabolic imaging." Proceedings of the National Academy of Sciences of the United States of America **103**(30): 11270-11275.
- Golman, K., L. E. Olsson, O. Axelsson, S. Mansson, M. Karlsson and J. S. Petersson (2003). "Molecular imaging using hyperpolarized ¹³C." Br J Radiol **76 Spec No 2**: S118-127.
- Hersman, F. W., I. C. Ruset, S. Ketel, I. Muradian, S. D. Covrig, J. Distelbrink, W. Porter, D. Watt, J. Ketel, J. Brackett, A. Hope and S. Patz (2008). "Large production system for hyperpolarized ¹²⁹Xe for human lung imaging studies." Acad Radiol **15**(6): 683-692.
- James, M. L. and S. S. Gambhir (2012). "A molecular imaging primer: modalities, imaging agents, and applications." Physiol Rev **92**(2): 897-965.
- Kurhanewicz, J., D. B. Vigneron, K. Brindle, E. Y. Chekmenev, A. Comment, C. H. Cunningham, R. J. DeBerardinis, G. G. Green, M. O. Leach, S. S. Rajan, R. R. Rizi, B. D. Ross, W. S. Warren and C. R. Malloy (2011). "Analysis of Cancer Metabolism by Imaging Hyperpolarized Nuclei: Prospects for Translation to Clinical Research." Neoplasia **13**(2): 81-97.
- Leermakers, P. A. and G. F. Vesley (1963). "Photochemistry of Alpha-Keto Acids and Alpha-Keto Esters .1. Photolysis of Pyruvic Acid and Benzoylformic Acid." Journal of the American Chemical Society **85**(23): 3776-&.
- Limes, M. E., Z. L. Ma, E. G. Sorte and B. Saam (2016). "Robust solid Xe-129 longitudinal relaxation times." Physical Review B **94**(9).
- Mansson, S., E. Johannesson, P. Magnusson, C. M. Chai, G. Hansson, J. S. Petersson, F. Stahlberg and K. Golman (2006). "¹³C imaging-a new diagnostic platform." Eur Radiol **16**(1): 57-67.

Marco-Rius, I., M. C. D. Tayler, M. I. Kettunen, T. J. Larkin, K. N. Timm, E. M. Serrao, T. B. Rodrigues, G. Pileio, J. H. Ardenkjaer-Larsen, M. H. Levitt and K. M. Brindle (2013). "Hyperpolarized singlet lifetimes of pyruvate in human blood and in the mouse." NMR Biomed **26**(12): 1696-1704.

Nelson, S. J., J. Kurhanewicz, D. B. Vigneron, P. E. Larson, A. L. Harzstark, M. Ferrone, M. van Criekinge, J. W. Chang, R. Bok, I. Park, G. Reed, L. Carvajal, E. J. Small, P. Munster, V. K. Weinberg, J. H. Ardenkjaer-Larsen, A. P. Chen, R. E. Hurd, L. I. Odegardstuen, F. J. Robb, J. Tropp and J. A. Murray (2013). "Metabolic Imaging of Patients with Prostate Cancer Using Hyperpolarized [1-13C]Pyruvate." Sci Transl Med **5**(198): 198ra108.

Nikolaou, P., B. M. Goodson and E. Y. Chekmenev (2015). "NMR Hyperpolarization Techniques for Biomedicine." Chemistry-a European Journal **21**(8): 3156-3166.

Vasos, P. R., A. Comment, R. Sarkar, P. Ahuja, S. Jannin, J. P. Ansermet, J. A. Konter, P. Hautle, B. van den Brandt and G. Bodenhausen (2009). "Long-lived states to sustain hyperpolarized magnetization." Proceedings of the National Academy of Sciences of the United States of America **106**(44): 18469-18473.

Warren, W. S., E. Jenista, R. T. Branca and X. Chen (2009). "Increasing Hyperpolarized Spin Lifetimes Through True Singlet Eigenstates." Science **323**(5922): 1711-1714.

Witte, C. and L. Schroder (2013). "NMR of hyperpolarised probes." Nmr in Biomedicine **26**(7): 788-802.

Yoshihara, H. A., E. Can, M. Karlsson, M. H. Lerche, J. Schwitter and A. Comment (2016). "High-field dissolution dynamic nuclear polarization of [1-(13)C]pyruvic acid." Phys Chem Chem Phys **18**(18): 12409-12413.

REVIEWERS' COMMENTS:

Reviewer #2 (Remarks to the Author):

All comments/suggestions have been addressed.

Reviewer #3 (Remarks to the Author):

You did a great job. Now I think the paper should be published as it is.